# Machine-learning predicts genomic determinants of meiosis-driven structural variation in a eukaryotic pathogen

Thomas Badet[1], Simone Fouché[1,2], Fanny E. Hartmann[3], Marcello Zala[2] & Daniel Croll [1✉]

Species harbor extensive structural variation underpinning recent adaptive evolution. However, the causality between genomic features and the induction of new rearrangements is poorly established. Here, we analyze a global set of telomere-to-telomere genome assemblies of a fungal pathogen of wheat to establish a nucleotide-level map of structural variation. We show that the recent emergence of pesticide resistance has been disproportionally driven by rearrangements. We use machine learning to train a model on structural variation events based on 30 chromosomal sequence features. We show that base composition and gene density are the major determinants of structural variation. Retrotransposons explain most inversion, indel and duplication events. We apply our model to *Arabidopsis thaliana* and show that our approach extends to more complex genomes. Finally, we analyze complete genomes of haploid offspring in a four-generation pedigree. Meiotic crossover locations are enriched for new rearrangements consistent with crossovers being mutational hotspots. The model trained on species-wide structural variation accurately predicts the position of >74% of newly generated variants along the pedigree. The predictive power highlights causality between specific sequence features and the induction of chromosomal rearrangements. Our work demonstrates that training sequence-derived models can accurately identify regions of intrinsic DNA instability in eukaryotic genomes.

[1] Laboratory of Evolutionary Genetics, Institute of Biology, University of Neuchâtel, Neuchâtel, Switzerland. [2] Plant Pathology, Institute of Integrative Biology, ETH Zurich, Zurich, Switzerland. [3] Ecologie Systématique Evolution, Bâtiment 360, Univ. Paris-Sud, AgroParisTech, CNRS, Université Paris-Saclay, Orsay, France. ✉email: daniel.croll@unine.ch

Structural variation including duplications, inversions, insertions, and deletions is often the most common form of genetic variation within species[1–3]. A wide range of phenotypes including human diseases, traits for crop improvements, and pathogen virulence are associated with chromosomal rearrangements[4–7]. Breaks in genome collinearity can have important consequences for the faithful exchange of alleles during meiosis[8–10]. Hence, chromosomal rearrangements can also underpin adaptation stemming from co-adapted alleles including sex determination loci[11,12]. Structural variation arises either from non-allelic homologous recombination, homology-directed repair, and microhomology-mediated end joining between repetitive regions in the genome or the activity of transposable elements[13,14]. However, predicting the likelihood of sequence rearrangement events remains challenging. In most species, the repetitive sequence content is poorly resolved at the nucleotide level leading to fuzzy predictions about fragile sites that are prone to rearrangements[15]. Furthermore, analyzing segregating structural variants at the population level only partially reflects sequence rearrangement events because selection tends to remove deleterious variants over generations.

Most sequence rearrangements are generated through double-strand break repair[16]. Germline double-strand breaks are typically induced during meiotic recombination[1]. Microhomology, tandem repeats, and large-scale duplicated sequences can be at the origin of fragile sites and cause chromosomal instability[17,18]. However, whether and how individual genetic and epigenetic sequence features attract higher rates of sequence rearrangements is poorly understood[16,19]. Pangenome studies across eukaryotes showed that structural variation segregating within species can exceed single-nucleotide variation[3]. Species-wide structural variation is often heterogeneously distributed along chromosomes and shows strong associations with epigenetic marks and the activity of transposable elements[9,20–22]. Purifying selection in populations removes deleterious chromosomal rearrangements over generations, hence skewing the spectrum towards neutral or nearly neutral structural variation[8].

Here, we establish a nucleotide-level map of structural variation based on 19 chromosome-level assemblies of the fungal pathogen *Zymoseptoria tritici*, which is one of the most devastating pathogens of wheat, causing 5–10% annual losses in Europe alone[23]. Based on genome-wide association mapping of 24 phenotypes characterizing the life cycle of the pathogen, we show that top associated SNPs tend to map near sequence rearrangement loci and that the recent emergence of pesticide resistance was disproportionally driven by structural variation. We apply machine learning integrating 30 sequence metrics to train a species-wide model to predict causal factors underlying structural variation. We show that the model has high precision and accuracy to predict insertion/deletion (indel) events and we successfully apply the same approach to predict structural variation in the pangenome of *Arabidopsis thaliana*. To disentangle the effect of purifying selection from the mechanisms promoting structural variation, we generate chromosome-level assemblies of nine haploid progeny over four generations. We assess the power of the pangenome-derived model and quantify the impact of parental background on the generation of sequence rearrangements.

## Results

**Pangenome-wide collinearity analyses reveal hotspots of structural variation.** To build a model of how structural variation arises within a species, we analyzed the high-quality and world-wide distributed pangenome of the fungal wheat pathogen *Z. tritici* (Fig. 1A). Homologous chromosomes show substantial breaks in collinearity across the global species range driven both by indels and inversions (Fig. 1B). Eight of the 21 chromosomes are accessory in the species with presence/absence polymorphism and can undergo drastic rearrangements such as the fusion of two chromosomes in the isolate from Yemen (Fig. 1B, C).

We performed all possible pairwise alignments between the 19 complete genome assemblies to analyze whether the extent of structural variation within the species varied among chromosomal regions. Collinearity (or synteny) can be expressed as a ratio of syntenic segments over the total number of syntenic and non-syntenic segments across windows (Fig. 1D). We averaged synteny scores across the pangenome and found distinct profiles along chromosomes. Telomeric regions were universally of low synteny consistent with higher densities of repetitive sequences and more opportunity for non-allelic homologous recombination. Core chromosomes showed marked drops in synteny in regions of major sequence rearrangements including large inversions on chromosomes 3 and 12. The inversion on chromosome 12 is present in 3 isolates (1E4, CRI10, and KE94), encompasses nearly 3% of the total chromosome length, and affects 11–17 genes (Fig. 1B). Accessory chromosomes showed greatly reduced synteny scores compared to core chromosomes with pronounced signatures of sequence dissimilarity. Overall, 40% of the species gene content is not fixed (i.e., accessory genes). Chromosomal length variation was positively correlated with transposable element content (Fig. 1E). Essential (core) chromosomes showed a cap at ~6% size variation despite significant variability in transposable elements. Accessory chromosomes showed significantly relaxed constraints on chromosome length and transposable element content.

**Long-read and whole-genome alignment-based structural variant analyses.** Local synteny scores do not resolve precise sequence rearrangement events. Hence, we analyzed structural rearrangements using PacBio long reads mapped against the reference genome (IPO323). Evaluating evidence from split mapped reads, mismatch, and coverage data, we identified a total of 21,718 structural variants localized at base-pair resolution. Detected variants included duplications (0.85%), indels (56.6%), inversions (0.95%), and translocations (41.6%) (Supplementary Data 1). To avoid false positives due to noisy long-read mapping, we analyzed sequence rearrangements using an alternative method based on pairwise whole-genome alignments. This second approach revealed a total of 50,767 unique rearrangements including duplications (5.3%), copy polymorphisms (6.5%), highly diverged regions (i.e., unresolved complex rearrangements) (16.2%), indels (52.5%), inversions (0.9%), inverted duplications (4.9%), inverted translocations (5.8%), tandem repeats (2.2%), and translocations (5.4%) (Supplementary Data 2). We consolidated variants based on physical proximity and type (see "Methods" section). After merging, both variant mapping methods yielded a similar number of structural rearrangements with 13,902 rearrangements overlapping between the PacBio read and whole-genome alignment methods (Supplementary Fig. 1). The pairwise whole-genome alignment approach revealed a more diverse set of structural rearrangements compared to the read mapping method (Supplementary Data 1 and 2). Read mapping is sensitive to read orientation and coverage at breakpoint positions, which likely explains the higher proportion of coarsely classified variants as translocations given unresolved breakpoints[24].

We performed resampling over the included genomes to assess the breadth of total structural variation within the species captured in our analyses. Both read mapping and genome alignment-based structural variation calling revealed a near-linear relationship between the number of genomes included and the

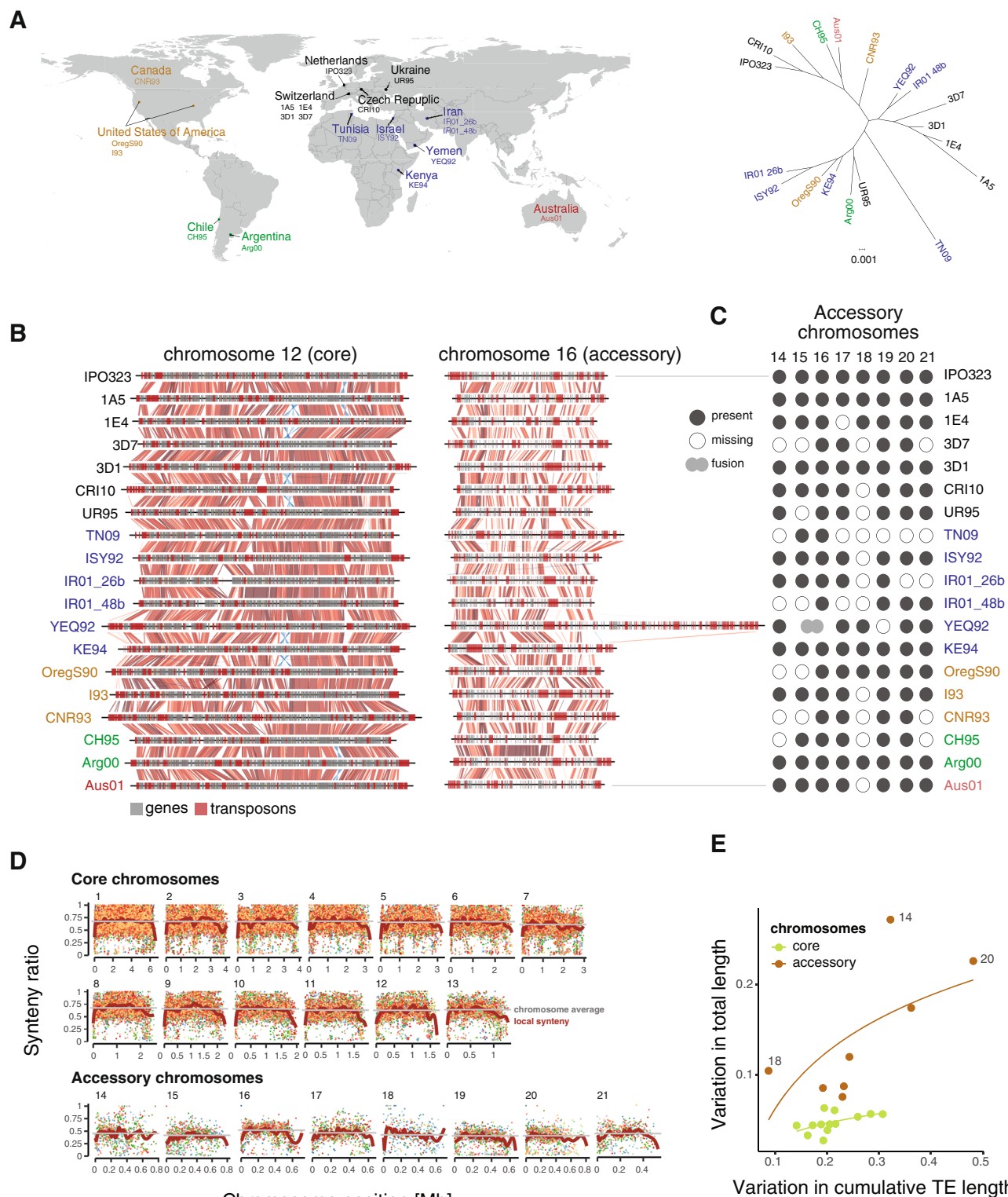

**Fig. 1 Species-wide genome structure of a highly polymorphic microbial eukaryote. A** Worldwide map showing the country of origin of the 19 *Zymoseptoria tritici* isolates and phylogenetic relationships based on 50 random orthologous genes. Colors indicate the continent of origin. The map was generated using the R package rworldmap. **B** Synteny plot of core chromosome 12 and accessory chromosome 16. The red and blue gradients represent the percentage of identity given BLAST alignments for syntenic and inverted segments, respectively. Darker colors show higher identity values. Genes and transposons are represented by gray and red boxes, respectively. **C** Presence/absence variation of accessory chromosomes. The isolate from Yemen (YEQ92) carries a hybrid chromosome following a fusion between chromosomes 15 and 16. **D** Genome-wide synteny ratios along chromosomes of the 19 isolates. Synteny ratios are shown for each isolate as represented by the colored dots. **E** Variation in the total transposable element (TE) content per chromosome. Core and accessory chromosomes are shown in green and brown, respectively. Length variation is expressed as the coefficient of variation. Curves show the best logarithmic fit.

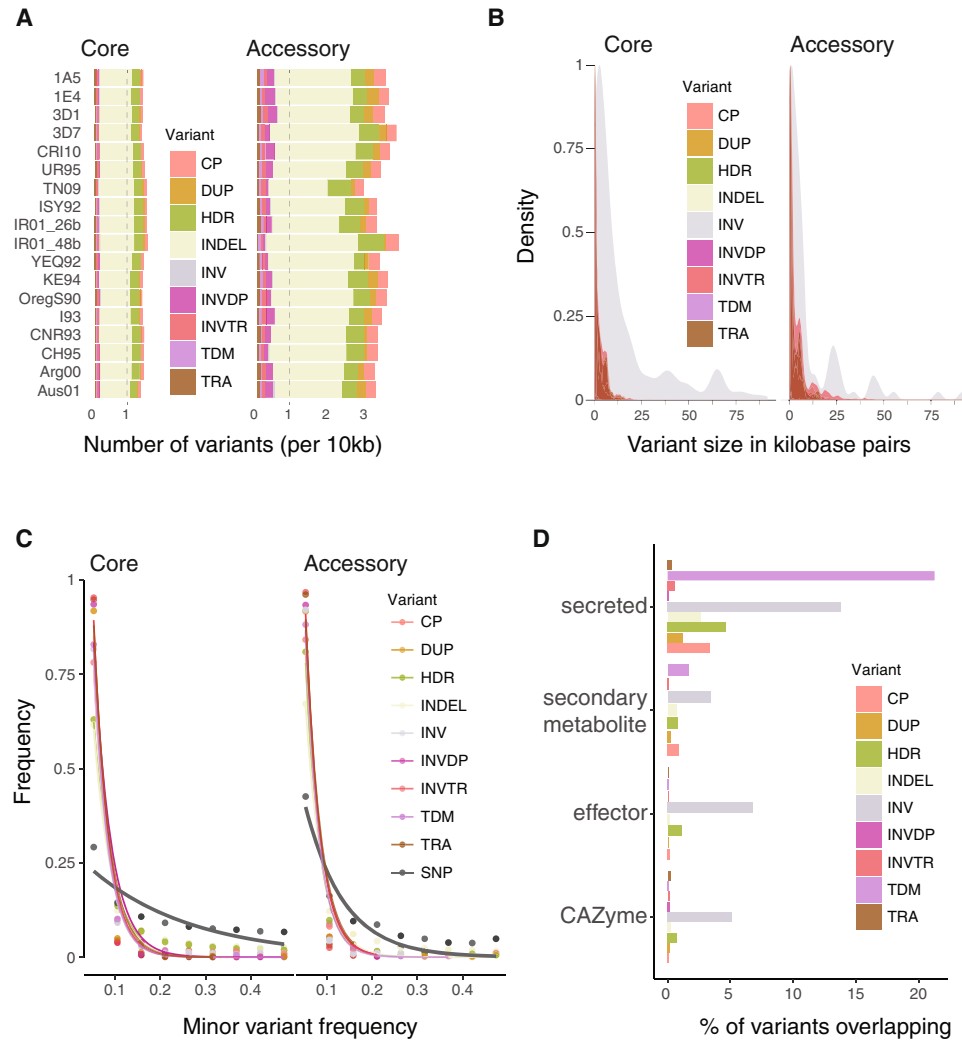

**Fig. 2 Structural variant heterogeneity across chromosomes and associations with genes. A** Whole-genome alignment-based structural variants per 10 kilobase pairs (kb) among isolates. Copy polymorphisms (CP), duplications (DUP), highly diverged regions (HDR), indels (INDEL), inversions (INV), inverted duplications (INVDP), inverted translocations (INVTR), tandem repeats (TDM), and translocations (TRA). **B** Size distribution of the nine structural variant types. **C** Minor allele frequency distribution. Curves show estimates of the exponential decay calculated for each variant type. **D** Proportion of each type of variant overlapping with different gene category playing a role in fungal pathogenesis including secondary metabolite clusters, genes encoding for predicted secreted proteins, effectors, and carbohydrate-active enzymes (CAZymes).

total number of variants identified (Supplementary Fig. 2). *Z. tritici* is among the most polymorphic microbial eukaryotes analyzed to date[25]. The geographic breadth of the 19 included isolates covers all major wheat-producing areas of the world. Hence, despite covering only a fraction of the total structural variation within the species, the detected variation should be representative of the total diversity.

For the following analyses, we focus on variants identified using the whole-genome alignment method given the higher confidence and breadth of the variant discovery. Core chromosomes bear on average fewer structural variants compared to accessory chromosomes but show a higher proportion of indels, highly disordered regions, and translocation events relative to other structural variant types (Fig. 2A). Most indels (~87%) are short (<100 bp) and only 0.4% are larger than 10 kb (115 variants; Fig. 2B). In contrast, most duplications and inverted duplications (96%), copy polymorphisms (89%), highly diverged regions (95%), translocations (96%), inverted translocations (94%) and tandem repeats (97%) are larger than 100 bp. In particular, ~34% of inversions are longer than 10 kb. We found that all types of rearrangements show a pronounced shift towards low allele

frequencies compared to genome-wide SNPs consistent with purifying selection (Fig. 2C).

**Structural variation underpinning fitness-related trait expression**. Major phenotypic innovations originate from chromosomal rearrangements[10,26,27]. Duplications and deletions affecting coding sequences can promote phenotypic change. We show that ~30% of the genome is affected by structural variation and that ~21% of the rearrangements overlap with coding sequences (Supplementary Fig. 3). Inversions are balanced rearrangements that preserve sequences and thus gene content. Consistent with this, we find that inversions are more likely to contain coding sequences (~57%; Supplementary Fig. 3). Genes encoding effector proteins with likely functions in host manipulation and genes encoding carbohydrate-degrading enzymes are overall less affected by structural variation (Fig. 2D). Nevertheless, the telomeric effector gene *AvrStb6* known as the major genetic factor controlling disease development on a broad range of wheat cultivars is affected by a local inversion polymorphism (Supplementary Fig. 3)[28]. The subtelomeric region harboring *AvrStb6* shows a

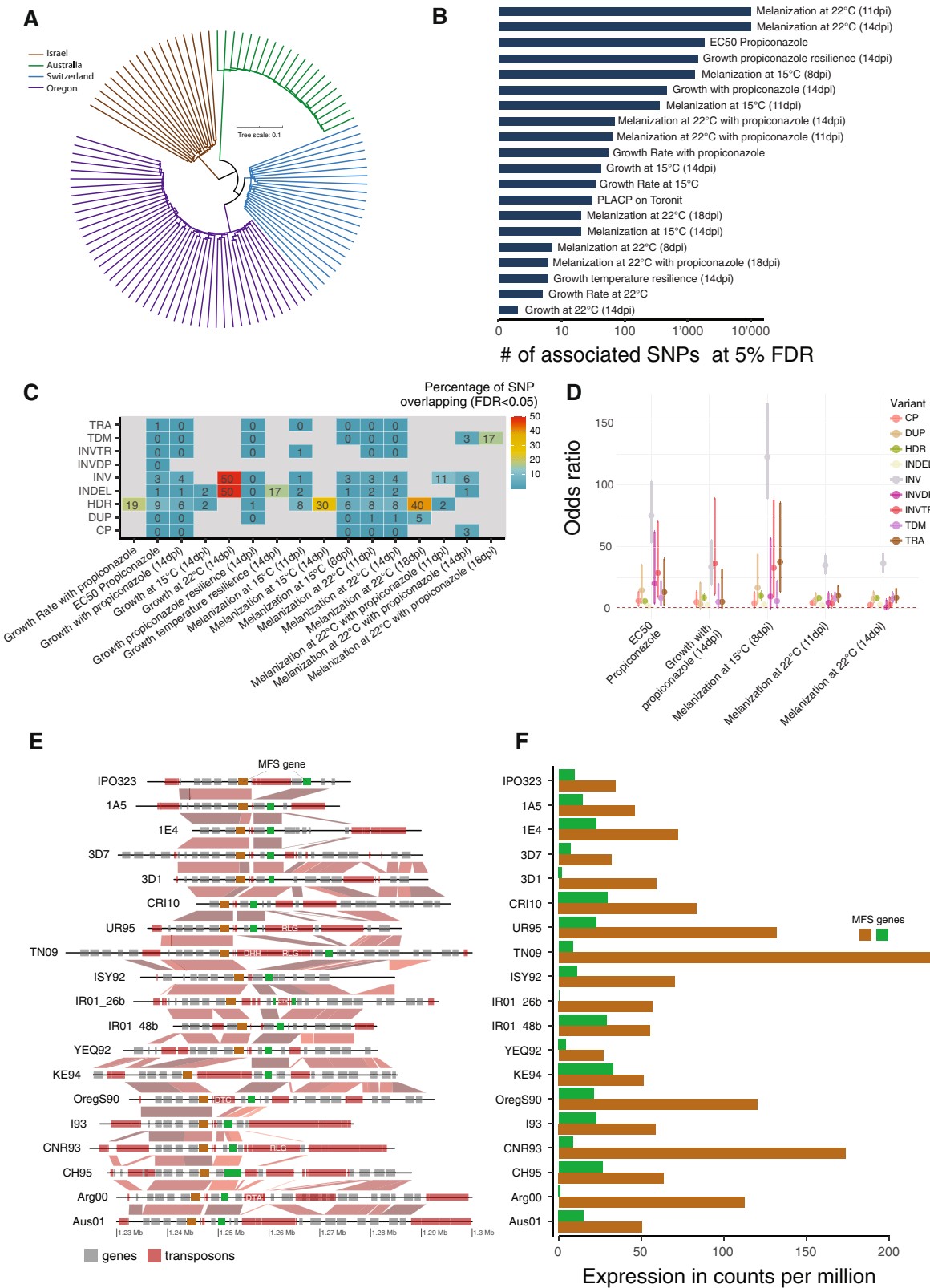

length variation of 30–250 kb and is one of the top 5% most rearranged chromosomal niches (Supplementary Fig. 4).

To quantify the contribution of structural variation to phenotypic trait variation, we performed genome-wide association analyses on 24 traits representative of the life cycle of the pathogen (Supplementary Data 3). The phenotypes included virulence on different wheat cultivars, temperature resilience, and

fungicide resistance. The mapping population consisted of 106 isolates from all major wheat-producing areas[29] (Fig. 3A). The analyzed polymorphisms explained between 0 and 26% of the phenotypic trait variation (for growth at 14 days after treatment with propiconazole and growth at 22 °C after 14 days, respectively; Supplementary Table 1). We identified 16 traits with significant SNPs located in regions affected by structural

**Fig. 3 Structural variation drives phenotypic trait evolution. A** Unrooted phylogeny of 106 isolates used for genome-wide association mapping of 24 traits resulting from the unweighted pair grouping method based on the arithmetic mean. **B** Number of significantly associated single-nucleotide polymorphisms (SNPs) across phenotypic traits (5% false discovery rate; FDR). **C** Percentage of significantly associated SNPs overlapping with structural variants across different phenotypes. **D** Odds ratio of associated SNPs overlapping structural variants. Bars represent the 95% confidence interval of two-sided Fisher's exact tests ($n = 779,178$ SNPs). Ratios are shown for five phenotypic traits associated with fungicide resistance including colony melanization. **E** Synteny plot of a locus associated with fungicide resistance on chromosome 13. Higher synteny between genomes is depicted by darker colored links. Two genes encoding major facilitator superfamily (MFS) genes are colored in brown and green. Nearby transposable elements are annotated according to their superfamily (shown in white). **F** Relative expression, expressed as counts per million of reads from one RNA sequencing experiment, of the two major facilitator superfamily (MFS) genes, coloration as in **E**.

variation. For melanin production, >10% of the associated SNPs overlap with regions with duplications, inversions, copy polymorphisms, highly diverged regions, indels, and tandem repeat variation (Fig. 3B, C). Additionally, we identified 1822 SNPs associated with tolerance to propiconazole, a widely used fungicide of the demethylation inhibiting (DMI) family. A total of ~14% of fungicide tolerance SNPs overlap with regions with indel variation, translocations, inversions, and highly disordered regions. Significantly associated SNPs were over-represented at sequence inversions, translocations, and inverted translocations for propiconazole tolerance and melanization-related traits (odds ratio > 2; Fig. 3D). Two fungicide tolerance SNPs were near major facilitator superfamily (MFS) genes. One of the MFS genes was also affected by a transposable element insertion and a loss of expression (Fig. 3F). Multiple additional insertions near the two MFS genes are associated with transcriptional variation (Fig. 3E, F). Taken together, regions underlying major phenotypic trait variation undergo high rates of structural variation.

**Chromosomal niches of structural variation hotspots.** Structural variation shows a highly heterogeneous distribution along chromosomes. Hence, specific sequence features may contribute disproportionally to the likelihood of rearrangements. In order to systematically identify associations across the genome, we analyzed variation in gene and transposable element densities, locations of core and accessory genes, isochores (i.e., AT-rich regions), recombination rates, and histone methylation marks for a total of 35 different sequence metrics available for the IPO323 reference genome across 10 kb non-overlapping windows. Gene and transposon coverage are strongly negatively correlated and underlie strong genome compartmentalization (Fig. 4A and Supplementary Data 4). The highly abundant transposable element superfamily *Gypsy* shows the strongest correlation with transposable element regions. Transposable elements co-localize both with repressive and facultative heterochromatin marks (H3K9$_{met3}$ and H3K27$_{met3}$, respectively). We find that all types of structural variation (except indels) are over-represented near (<10 kb distance) of transposable elements. The strongest over-representation was found for translocations and inverted translocations (Fig. 4A and Supplementary Fig. 5). This shows that transposable elements are drivers of structural variation in specific compartments of the genome. Indels show no preferential associations across the genome but tend to be more abundant near (<10 kb distance) accessory genes and depleted near (<10 kb distance) transposons. Repressive heterochromatin marks and transposable element density (*Gypsy* elements in particular) are strongly predictive of translocations, inverted translocations, duplications, and inverted duplications. Overall, chromosomal rearrangement hotspots are associated with H3K9 and H3K27 histone methylation marks, specific transposable elements, and low gene density.

**A pangenome-informed model accurately predicts structural variation.** The species-wide structural variation is the joint product of intrinsic instability of specific sequence features and selection acting on structural variants. To build a predictive model accounting for these different forces, we integrated 30 sequence features available for all complete genomes including the progeny dataset. Features included the density and identity of transposable elements, the presence of virulence factors, local recombination rates, GC content as well as information on gene dispensability (see Supplementary Data 5 for metric details). Even though gene dispensability is itself potentially an indicator of structural variation, the majority of dispensable genes in *Z. tritici* were generated by loss-of-function mutations[30]. We evaluated four different machine-learning algorithms with three times 10-fold cross-validations. We selected the best models based on accuracy and maximized the percentage of correctly classified instances overall. Overall, the four algorithms performed similarly, yielding 0.84 average accuracy across the nine types of rearrangements (Fig. 4B, C). The random forest method yielded slightly higher precision (0.64) while the generalized boosted regression method achieved better recall and F1 statistics on average (0.64 and 0.77, respectively) (Supplementary Fig. 6 and Supplementary Data 6). For all types of rearrangements and the random forest method, the best predictors included GC content, recombination rate, the presence of core genes, and transposable elements (Supplementary Fig. 7). Weaker predictors included the presence of retrotransposons for duplications, inversions, and translocations (Supplementary Fig. 7). Models for inversions and tandem duplications, the two rarest type of variants yielded high accuracy (due to true negative detection) but had low precision and recall (no true positives detected). Given the measures of area under the receiver operating characteristic (auc-ROC) curve and precision-recall (auc-PROC), the best predictions were achieved for indels (auc-roc = 0.69; auc-proc = 0.99) and highly diverged regions (auc-ROC = 0.69; auc-PROC = 0.74) (Fig. 4B, C and Supplementary Data 7). Both indels and highly diverged regions also yielded the highest number of true positives (Fig. 4E; Supplementary Data 8). However, the final models for highly diverged regions tended to call both false positive and false negative variants (~15% and 18%, respectively).

We replicated our approach to predict structural variation in the model plant *Arabidopsis thaliana*. The pangenome shows hotspots of rearrangements associated with transposable elements and depleted for recombination[31]. We parametrized the random forests model based on Col-0 reference sequence metrics including transposable elements, core, and accessory gene coverage, recombination rate, and nucleotide-binding and leucine-rich repeats (NLRs) gene coverage[31,32]. Over the total of 3974 sequence windows, we correctly predicted 100% of the indel polymorphism ($n = 3,947$) with an accuracy of 0.99 and a recall of 1 (Fig. 4F and Supplementary Table 2). Additionally, we correctly predicted 26% of the highly diverged regions, 20% of the duplications and copy polymorphisms, 17% of the inverted duplications and translocations, and 16% of the inverted translocations (Supplementary Table 3). The models were not able to predict the two rarest variants present in the species,

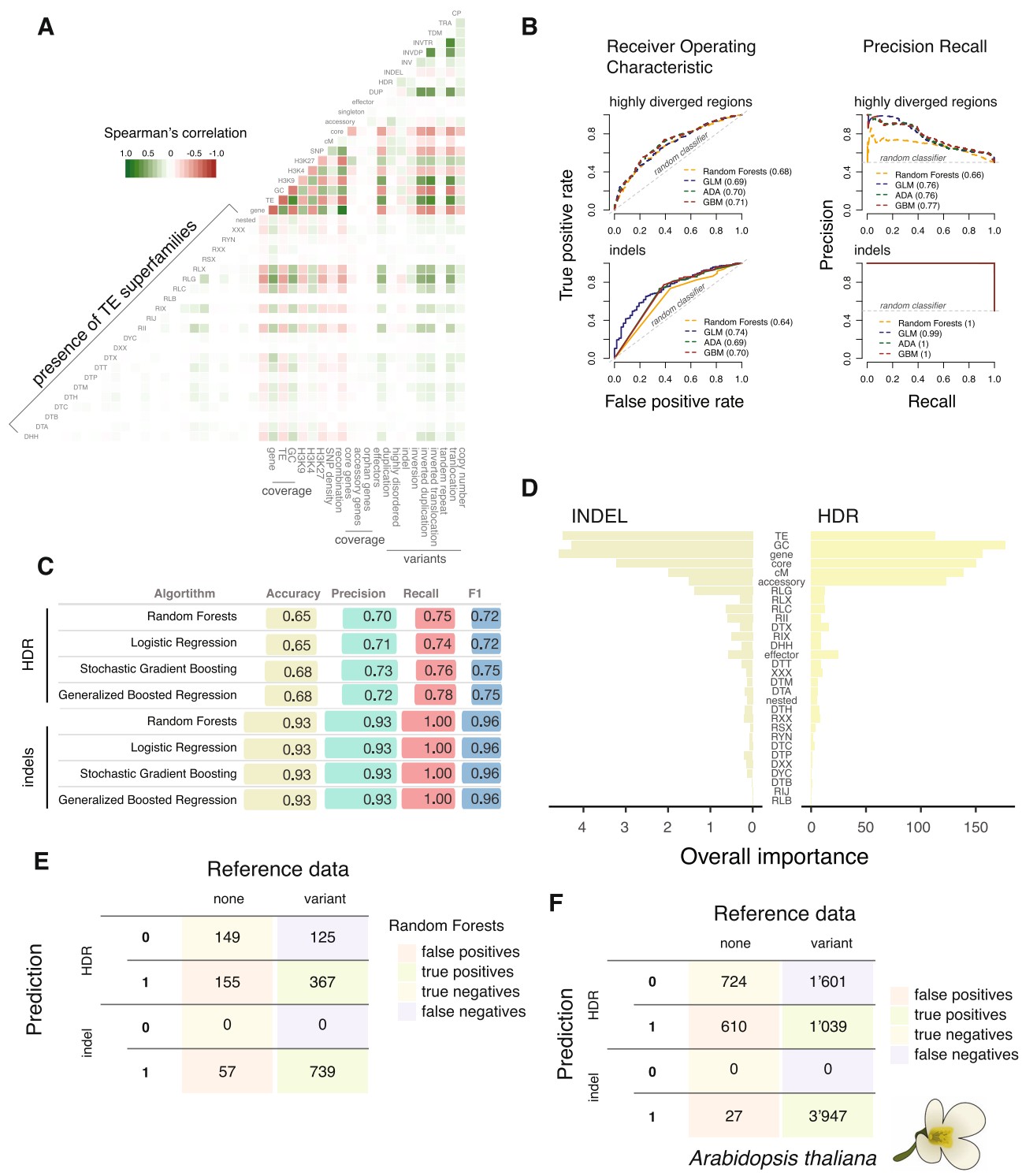

**Fig. 4 A pangenome-informed model predicts the occurrence of structural variation. A** Correlation analyses of 35 genome-wide metrics available for the reference genome IPO323 with types of structural rearrangements. Correlations were assessed from pairwise complete observations. **B** Receiving operating characteristics and precision-recall curves of the final models trained using highly diverged regions (upper part) and indels (lower part). Four algorithms were analyzed including random forests, logistic regression (GLM), stochastic gradient boosting (ADA), and generalized boosted regression (GBM). Numbers in parentheses indicate the area under the curve for each of the four models. **C** Summary statistics of the final models for translocations and indels using the four algorithms. Accuracy describes the model capacity at only detecting true variants. Precision measures the ability to predict true variants. Recall measures the ability to detect all true variants and the F1 score considers the harmonic mean of both precision and recall. **D** Relative importance of each of the 30 sequence features used in the training of the random forests model. The importance of individual sequence features is shown as the unscaled averages of the class-specific scores. **E** Confusion matrix resulting from highly diverged regions and indel predictions using the two respective models trained with the random forests algorithm. **F** Confusion matrix of the highly diverged regions and indel predictions performed for the *Arabidopsis thaliana* pangenome.

namely inversions and tandem duplications (122 and 536 for *A. thaliana*, respectively). The successful application of machine learning to predict structural variation suggests that eukaryotic genomes rearrange based on identifiable patterns in genome sequences.

**The predictive power of the species-wide model in an experimental pedigree.** Structural variation within species is likely produced de novo during meiosis or accumulate in somatic tissue. Nearly all observed variants are expected to be neutral or nearly neutral because of purifying selection. To experimentally investigate mechanisms underlying the formation of new rearrangements, we produced a haploid pedigree of four generations of experimentally crossed isolates (Fig. 5A). The parental isolates were collected from Swiss wheat fields and are included in the species pangenome analyses. We crossed the isolates using an established protocol to produce sexual progeny on inoculated wheat leaves. Sexual progenies were harvested and sub-cultured as single genotypes for further experiments. The haploid stage is dominant in *Z. tritici* meaning that any progeny has already undergone meiotic recombination of the two haploid parental genomes. Hence, chromosomal rearrangements triggered by sexual reproduction can be scored directly in an F1. We used progeny and parental strains to generate a total of five crosses with the first one being between the two parents and four subsequent backcrosses with one of the two parents (Fig. 5A). The total of 9 progeny was separated each by one to four rounds of meiosis from the parents (see "Methods" section). We produced gapless, chromosome-scale assemblies using PacBio sequencing. The genome assemblies revealed that the accessory chromosome 17 displayed significant instability undergoing large segmental duplications (Supplementary Fig. 8). Four accessory chromosomes were lost after the third and fourth round of meiosis likely due to non-disjunction events (Fig. 5C). Progeny genomes showed large drops in synteny compared to parental isolate 1A5 consistent with the large inversions segregating among the parental genomes (Supplementary Figs. 9 and 10). Regions of low synteny among progeny genomes were largely congruent with low synteny regions in the species pangenome, suggesting that the same sequences promote recurring structural rearrangements. Following the procedures for structural variant calling to train the species-wide model, we identified rearrangements in progeny genomes by aligning the nine progeny whole-genomes against one parental genome (i.e., 1A5). We filtered out any structural variant segregating between the two parental genomes. This ensured that we focused on structural variants that have originated within the pedigree. We identified a total of 1464 de novo rearrangements including 363 indels, 106 highly diverged regions, 380 inverted duplications, 283 duplications, 100 copy polymorphisms, 111 inverted translocations, 77 translocations, 38 tandem repeats, and 6 inversions matching the distribution and size observed in the species-wide genome collection (Fig. 5B and Supplementary Data 9). We found that the initial cross 1A5 × 1E4 (first meiotic event, M1) has a higher rate of new rearrangements when compared to the three other meiotic rounds in the pedigree (interaction test for meiotic round and variant type; two-way analysis of variance, $p$-value $< 1e^{-6}$). These observations are consistent with a role for parental genome dissimilarity in promoting rearrangements. We conclude that the few rounds of meiosis in the pedigree were sufficient to reproduce major aspects of the species-wide set of chromosomal rearrangements.

Our design of crosses and backcrosses enables us to retrace the emergence of structural variants through individual rounds of meiosis (Fig. 5D). To dissect the history of new rearrangements along the crosses, we assigned each variant to one of the four categories. (1) New variants that were generated in the most recent round of meiosis. Overall, isolates resulting from a backcross with the 1E4 parent tend to show higher numbers of new variants (Fig. 5B–D). The progeny isolates from the first meiosis cross with the 1A5 and 1E4 parents show the highest number of new structural variants (A2.2 and A66.2). (2) Lost variants compared to the parental genomes were most frequent in the first backcross as expected. (3) Retained variants shared by all progeny of a cross are rare and accumulate in the pedigree. (4) We detected recurring mutations in nearly all progeny beyond the second round of meiosis (i.e., identical variants generated multiple times independently). A total of 189 of the new progeny variants (including all types of rearrangements except inversions) are also present in the pangenome. New and lost variants tend to occur in distinct regions of the chromosome as expected from crossovers re-introducing parental sequences (Supplementary Fig. 11). We dissected the mechanism by which new and recurrent rearrangements are generated. For this, we identified crossovers from whole-genome alignments between each progeny and the two respective parental genomes (Fig. 5E). As expected, segments inherited from the backcrossed parental isolate were depleted of structural variants (Fig. 5E). The two transposable element superfamilies (RLG and RII elements) driving structural variation across the species showed no clear association with crossover events and new rearrangements in the pedigree. Interestingly, we accurately predicted an indel and a translocation event that occurred in the progeny near the two MFS genes associated with fungicide resistance in our genome-wide association study. New structural variants overall tend to be generated near crossover breakpoints (Fig. 5E). To retrace how crossover generates new rearrangements, we used synteny to localize crossover breakpoints on chromosome 1 through the 4 meiotic cycles ($n = 18$ breakpoints). We show that indels and inverted duplications are enriched in proximity to crossover breakpoints consistent with recombination-mediated double-strand breaks causing sequence rearrangements (Fig. 5F).

Sequence rearrangements generated through meiosis can impact fitness and therefore challenge the analyses of structural variation in natural populations. Progeny from our pedigree was minimally cultured on a nutrient-rich medium likely reducing the strength of selection on slightly deleterious rearrangements. To test the predictive power of the species-wide model of structural variation, we re-parametrized the input metrics for the one parental genome (i.e., 1A5) in the pedigree. Using the same set of sequence features across windows, we analyzed the occurrence of a total of 1464 newly generated rearrangements across 3994 genomic windows. As for the species-wide model, no algorithm clearly outperformed others in terms of accuracy, precision, and recall metrics (Supplementary Fig. 12). The highest F1 score metric was achieved for duplications and inverted duplications ($0.29 < F1 < 0.37$) with an accuracy ranging from 0.80-0.84. Both indels and highly diverged regions yielded lower accuracy compared to the species-wide model and the *Arabidopsis* model (Supplementary Fig. 12). However, using the random forests model, we were able to accurately predict 100% of the indels and 74% of the highly disordered regions generated in the pedigree (Fig. 5G). Results for duplications, copy polymorphisms, translocations together with inverted duplications and inverted translocation also yielded high proportions of accurately predicted events (79%, 77%, 94%, 81%, and 89%, respectively) (Supplementary Data 10). The lower accuracy of the model applied to the pedigree compared to the species-wide model stems from an over-prediction of structural rearrangements (Fig. 5G and Supplementary Table 4). We investigated the genomic context of the over-predictions using synteny ratios between the parental genomes. We find that true negatives tend

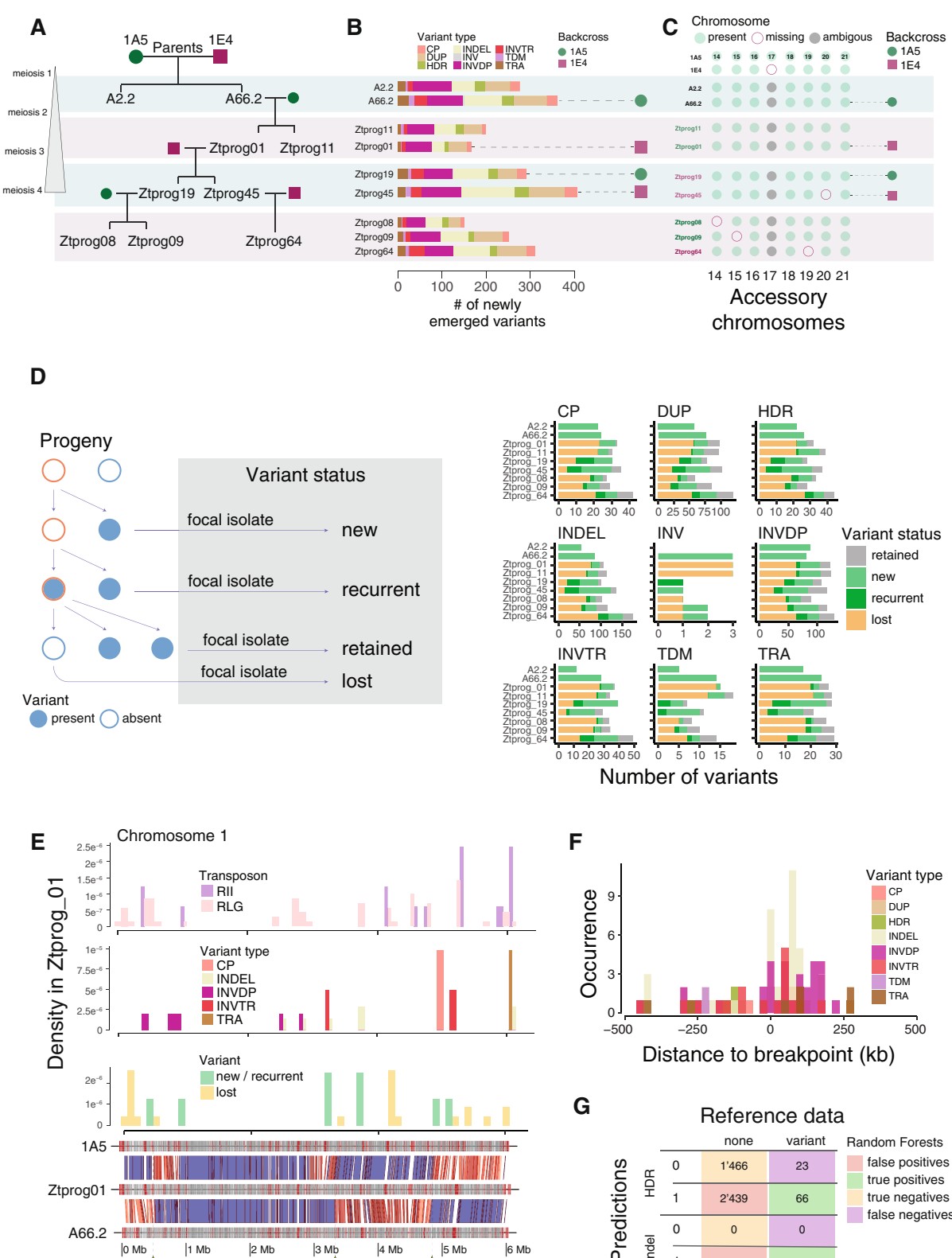

to be in regions of high synteny (Supplementary Fig. 13). On the contrary, true positives tend to be in regions of low synteny compared to false positives in particular for inverted duplications (Wilcoxon rank-sum test, $p$-value < 0.05). This suggests that model over-predictions are caused by variation in synteny between the two parental genomes (Supplementary Fig. 13). False positives are found in regions of higher recombination rate

compared to true positives (Wilcoxon rank-sum test, $p$-value < 0.0001). Consistent with this, we find that the model predicts less false positive within 5 kb of crossover events except for indels and highly diverged regions. In addition, the ratio of false and true discovery rates among progeny is independent of the round of meiosis in the pedigree (Supplementary Fig. 14). In general, model performance is lowest for the second round of meiosis with

**Fig. 5 Predictions of structural variation generated by meiosis throughout a pedigree. A** Pedigree chart showing the relationship between the 9 progeny isolates retrieved across four rounds of meiosis. Colored rounds and squares indicate the parental isolate used for backcrosses. **B** Number of newly emerged variants detected in each of the progeny isolates. Isolates are ordered as in the pedigree chart and colored rounds and squares indicate the parental isolate used for backcrosses. Copy polymorphisms (CP), duplications (DUP), highly diverged regions (HDR), indels (INDEL), inversions (INV), inverted duplications (INVDP), inverted translocations (INVTR), tandem repeats (TDM), and translocations (TRA). **C** Presence/absence of accessory chromosomes across the progeny set and the parental isolates. Chromosome 17 is absent in the 1E4 parent and shows complex segregation in the pedigree including truncations and duplications. **D** Classification of variants according to their origin and persistence in the pedigree. Variants present in a focal progeny and the direct parents were defined as "retained" variants. Variants present in a focal progeny but in none of its ancestors in the pedigree were called "new" variants. Variants present in a progeny, absent in its direct parents but present already in an ancestor were defined as "recurrent" variants. Finally, variants absent from a progeny but present in both parents were called "lost" variants. **E** Chromosome 1 synteny plot of the progeny Ztprog01 and its two parental isolates (lower panel). Regions of maximal pairwise identity (i.e., 100%) are colored in blue and highlight the three recombination events (green stars). Chromosome density of new and recurrent as opposed to lost structural variants (second panel from bottom). The number of new indels and translocations (second panel). The density of the two transposable elements contributing most to structural variation based on machine learning. RII: LINE I retrotransposons, RLG: *Gypsy* retrotransposons. **F** Density plot of new structural variants related to the closest crossover breakpoints on chromosome 1 expressed in kilobase pairs (kb). **G** Confusion matrix resulting from indel and highly diverged region model predictions using the two respective models trained with the random forests algorithm.

the 1A5 parent and increases towards the last round of meiosis (i.e., with an increase in detected variants). Altogether, we show that meiosis contributes significantly to the formation of structural variation in the species, and that machine learning enables the precise prediction of indel events across rounds of meiosis.

## Discussion

We show that integrating sequence data into an optimized model enables us to make powerful predictions where structural variation is generated along chromosomes. Similar sequence features underpin the likelihood of rearrangements both in the sets of species-wide genomes of *Z. tritici* and *A. thaliana*. The major factors leading to higher rates of sequence rearrangements are low GC and gene content. Low GC content is associated with a variety of sequence features including short repeats and inactivated transposable elements. The higher rate of sequence rearrangements triggered by these regions is likely a combined effect of relaxed selection and higher degrees of homology along the genome. Indeed, AT-rich regions are more likely to form secondary structures that disturb DNA replication[33]. Structural variation occurring in gene-dense regions is more likely to be deleterious. Here, we show that recent adaptation to resist pesticide applications has been primarily driven by structural variation. We identified rearrangements near two major facilitator superfamily (MFS) encoding transporters of xenobiotics. This shows that strong positive selection on individual rearrangements can exceed purifying selection acting on gene-rich, conserved regions.

Germline sequence rearrangements are mainly generated during meiosis. However, selection on deleterious rearrangements can potentially skew how population-level structural variation relates to underlying sequence features. The haploid fungus *Z. tritici* enables observing rearrangements generated by meiosis already at the F1 stage and growth on a rich medium minimizes the breadth of selection compared to natural environments. Meiotically induced structural variants are often the result of non-allelic homologous recombination[16]. Genome-wide structural variation breakpoints in the human genome revealed that such NAHR events are associated with higher recombination rates, but also higher GC content and open chromatin, suggesting conserved mechanisms across eukaryotes[18]. In *A. thaliana*, structural variation is negatively correlated with recombination rates likely due to structural variation being able to suppress crossover formation[9]. Here, we show that new variants are preferentially generated near crossovers showing that breakpoints can act as

hotspots for creating structural variation including recurring variants.

Our species-wide model of structural variation enables powerful predictions about the occurrence of sequence rearrangements during meiosis. Our pedigree analyses additionally revealed a significant number of recurrent rearrangements at relatively short distances, which were generated independently multiple times. This shows that chromosomal instability can be mapped precisely in regard to chromosomal sequence features. Recurrent and nonrecurrent structural rearrangements in humans are thought to be generated by different mechanisms. NAHR between low-copy repeats is a major factor for the recurrent mutations underlying genetic disorders in humans[34]. Similarly, recurrent deletions and insertions influence developmental processes in sticklebacks[35] and cell aggregation in yeasts[36], respectively. The ability to predict the likelihood and nature of structural variants from sequence features alone provides a conceptual ground for the growing evidence of parallel evolution at the molecular level[37,38]. Here, we show that indels and duplications are most likely generated through retrotransposon activity. Similarly, inversions are generated near low-copy LINE retrotransposons. In the progeny, these transposable elements showed no close association to the newly generated rearrangements but rather with crossover breakpoints. This suggests that ongoing lineage-specific repeat expansions are the primary driver of sequence rearrangements. Genomic models trained on species-wide structural variation will have powerful applications in research on genetic disorders, assessing the persistence of genetic engineering in crop breeding and the development of evolutionarily stable antimicrobials.

## Methods

**Experimental crossings.** We generated a four-generation pedigree started from a cross of the isolates 1A5 with 1E4 (PRJEB15648 and PRJEB20900, respectively) initially described in[39]. One haploid progeny was selected for further sexual crosses. The isolate A66.2 was subjected to further crosses and backcrosses with the parental isolates following the protocol described by[40]. The full pedigree is described in Fig. 5A. Crosses were made by co-inoculating spores of the parental pairs on wheat plants and incubated until the development of pseudothecia (sexual structures of the fungus). Ascospores ejected from pseudothecia were isolated on water agar plates. Germinating ascospores were transferred for clonal propagation and further DNA extraction.

**Species pangenome and pedigree analyses based on complete genomes.** The 19 isolate pangenome was first reported by Badet et al.[41]. Briefly, isolates were collected from six different continents on multiple wheat cultivars and span the climatic gradient of the species. High-molecular-weight DNA was extracted from cultured spores using a phenol–chloroform–isoamyl alcohol solution[30]. Libraries were prepared from at least 15 μg of DNA and the sequencing was performed using P6/C4 chemistry on a PacBio RSII or Sequel instrument at the Functional

Genomics Center, Zurich, Switzerland. PacBio reads were assembled to chromosome level using Canu v1.7.1 and Ragout v2.1.1 and polished twice using Arrow v2.2.2[42,43]. Genes were predicted using BRAKER v2.1 pipeline with RNA-seq and protein data for intron calling[44–51]. Orthogroups were defined based on protein homology using Orthofinder v2.1.2[52,53]. Transposable elements consensus sequences were identified de novo using RepeatModeler open-1.0.11 and annotated using RepeatMasker after manual curation[54]. We used the consensus sequences from Badet et al.[41] for repeat annotation. Briefly, repeats were identified de novo with RepeatModeler software using the 19 pangenome isolates together with the genome from the closest sister species Z. pseudotritici. Consensus sequences were manually curated and classified according to the GIRI Repbase[55]. Transposon superfamilies were named based on the three-letter classification system[56]. The genome assemblies from the 19 pangenome isolates and the 9 progeny isolates were annotated using the curated consensus sequences with RepeatMasker v4.1.0 software. The cut-off value was set to 250 and simple repeats, as well as low complexity regions, were ignored. We filtered out all the elements that were shorter than 100 bp. Adjacent identical elements that were overlapping by more than 100 bp were merged. Elements that were different and overlapping by more than 100 bp were considered as nested insertions and renamed accordingly. Interrupted elements were grouped into a single element using minimal start and maximal stop positions if separated by less than 200 bp. The world map presented in Fig. 1 was created with the R package rworldmap v_1.3-6.

The 9 progeny from the pedigree selected for genome sequencing were subjected to the same procedure as described above for DNA extraction and sequencing. Libraries were prepared from 15 to 31 μg of DNA after size selection with an 8 kb cut-off on a BluePippin system (Sage Science). The average fragment length was ~15 kb and further sequencing was performed using either P4/C2 or P6/C4 chemistry on a PacBio RSII or Sequel instrument at the Functional Genomics Center, Zurich, Switzerland. PacBio read assembly was performed using HGAP version 4 of the SMRTanalysis suite (version 6, release 6.0.0.47841)[57]. HGAP was run with the default parameters, except for the minimum seed read length, to initiate the self-correction. First, we produced assemblies for all chromosomes except chromosome 17 using the cutoffs automatically chosen by HGAP. In order to improve the contiguity of the assembly, we tested minimum seed read lengths of 8000, 10,000, 12,000, 15,000, 20,000. Chromosomes that could not be assembled into a single contig were scaffolded into chromosomes using Ragout (Reference-Assisted Genome Ordering UTility) version 2.2 with default parameters and Sibelia for synteny block decomposition[43,58]. Unusual assemblies were analyzed using Quiver 6.0.0.47835 with default settings as implemented in the SMRTanalysis suite and mapped reads were inspected visually in IGV (2.4.10). The synteny of assembled progenies was compared to the parent isolate 1A5 using Nucmer (mummer 3.23) with the parameter's delta-filter -i 80 -l 1000 -1 -q. A mummerplot was generated with the parameters–filter–large–fat–layout-t postscript to visually inspect assembly contiguity (Supplementary Fig. 8).

**Chromosome-wide synteny ratios.** To estimate the sequence collinearity between multiple chromosome-level assemblies, we adapted the method by Jiao et al.[31]. Briefly, we performed all-against-all whole-genome alignments of the 19 pangenome isolates using nucmer v4.0.0beta2 allowing for multiple matches (–maxmatch option). We filtered the alignments for a minimal identity of 90% and a minimal single match length 100 bp using the delta-filter in MUMmer v4.0.0beta2. We then extracted the alignment coordinates using show-coords with the -THrd options and mapped the position of syntenic and rearranged regions using the SyRi v1.3 software[24]. We calculated the intersect between the syntenic and rearranged regions given by SyRi and 10 kb windows along the 19 reference genomes using bedtools intersect (v2.29.2). The synteny ratio was then calculated as the number of syntenic blocks (named "SYN" and "SYNAL" in the SyRi output) over the number of total syntenic and non-syntenic blocks in each 10 kb window. For accessory chromosomes not shared by all isolates, the total number of chromosomes included in the synteny ratio was lower than for core chromosomes. For the pedigree analyses, we applied the same method to calculate the synteny ratio of the 9 progeny genomes with the exception that chromosome 17 was excluded from the analysis together with chromosomes 12 and 6 in the isolate Ztprog08. These chromosomes underwent major fusion events rendering the one-against-one alignments impossible.

**Whole-genome-based variant analysis.** To map structural variation across the 19 chromosome-level assemblies, we applied the method developed by Goel et al.[24]. Briefly, we performed whole-genome alignments of 18 genomes against the IPO323 reference genome using nucmer v4.0.0beta2 allowing for multiple matches (–maxmatch option). We filtered the alignments for a minimal identity of 90% and a minimal single match length of 100 bp using the delta-filter in MUMmer v4.0.0beta2. We then extracted the alignment coordinates using show-coords with the -THrd options and mapped the position of syntenic and rearranged regions using the SyRi v1.3 software[24]. For accessory chromosomes not shared by all isolates, the total number of chromosomes included for analysis was lower than for core chromosomes. We considered for further analysis the focal rearrangements (i.e., variants called for alignments between homologous chromosome pairs) annotated by SyRi as inverted (INV), translocated (TRA), inverted translocated (INVTR), duplicated (DUP), inverted duplicated (INVDP) and highly diverged

regions (HDR), together with insertions and deletions considered here as indels (INDEL); as well as copy gain and loss considered as copy polymorphisms (CP). Each variant location was given by the breakpoint positions in the IPO323 reference genome. The variant length was calculated as the difference between the two breakpoint positions in the IPO323 reference genome except for insertions, for which we considered the distance between the two breakpoint positions in the other genome. For all further analysis, we considered all individual variants larger than 10 bp but shorter than 100,000 bp.

**Long-read structural variant analysis.** Structural variants were identified following the pipeline described in ref.[59]. Briefly, PacBio reads from 18 isolates were aligned to the IPO323 reference genome using the NGMLR software v0.2.7 with default parameters. Variants longer than 10 bp (-l 10 option) were called using Sniffles v1.0.10 allowing for a maximum distance of 5 kb for grouping variants (-d 5000), a minimum mapping quality of 30 (-q 30) and a minimum read support of 80% (-f 0.8). Sniffles adds a tag to each variant reporting the confidence of the breakpoints mapping. We only retained precisely mapped variants that are smaller than 100 kb (translocation events have an unresolved length with the variant calling method). All 18 individual variant files were merged using the SURVIVOR v1.0.7 tool allowing a distance of 1 kb to bin variants[60]. Variants larger than 100,000 bp were filtered out for all further analyses. For comparisons between the long-read mapping and the whole-genome alignment method against the same IPO323 reference, we merged in both cases identical variant types separated by less than 1000 bp.

**Genome-wide association mapping.** The genome-wide association study (GWAS) was performed on 106 Z. tritici isolates collected from single wheat fields in Australia, Israel, Switzerland, and USA (Oregon) as described in[29]. The total SNP dataset included 779,178 SNPs. We analyzed phylogenetic relationships using the vk suite v0.2.8 (https://vcf-kit.readthedocs.io/en/latest/phylo/). We generated a phylogenetic tree using the vk phylo tree function using the unweighted pair grouping method based on the arithmetic mean. The phylogenetic tree was visualized with the iTOL web interface (https://itol.embl.de). We performed GWAS for a set of 24 traits related to fungicide resistance, growth, and host adaptation (for values see Supplementary Data 3). Virulence on two wheat cultivars was measured as the percentage of leaf area covered by pycnidia (PLACP) and described in ref.[29]. The other 22 traits were measured in vitro following methods from earlier studies[61–64]. Briefly, isolates were regenerated from −80 °C culture stocks on yeast malt sucrose agar Petri dishes for 4–5 days at 18 °C (4 g/L yeast extract, 4 g/L malt extract, 4 g/L sucrose, 50 mg/L kanamycin, 15 g/L agar). For plate inoculations, blastospores were diluted in sterile water to 200 spores/mL final concentration using KOVA counting slides and 500 μL spore suspensions were then spread on potato dextrose agar Petri dishes (PDA, 4 g/L potato starch, 20 g/L dextrose, 15 g/L agar). Control treatments were performed at 22 °C and cold treatment at 15 °C, both at 70 relative humidity. Plate pictures were taken at 8, 11, and 14 days post inoculation using a digital camera for five technical replicates. Images were analyzed using ImageJ macros from ref.[61] to measure colony area. For each isolate, estimates of colony area are based on nine single spore colonies. Growth rates (mm/day) were estimated from the average colony radius by fitting a generalized linear model over the three time points. Melanisation was estimated from the same images by calculating the average values of gray (colors ranging from 0/black to 255/white). For fungicide treatments, 0.05 ppm of propiconazole was added to the PDA plates to address its impact on growth at 15 and 22 °C. Further formal testing of propiconazole resistance was carried on microtiter plates. For that spores were grown on 100 μL Sabouraud-dextrose liquid media (inoculated with 100 μL of spore suspension at 2.5e⁴ spores/mL) with varying concentrations of propiconazole (0.00006, 0.00017, 0.0051, 0.0086, 0.015, 0.025, 0.042, 0.072, 0.20, 0.55, 1.5 mg/L including a control without propiconazole). Microtiter plates were sealed and incubated at 22 °C for four days at dark with 80% relative humidity. Growth was estimated given the optical density at 605 nm using an Elisa plate reader (MR5000, Dynatech). Five technical replicates were performed, and dose-response curves were used to estimate the concentration at which the growth is reduced by half ($EC_{50}$) using the drc v.3.0-1R package[65]. Prior to the association study, we computed the genetic relatedness matrix (GRM) and conducted a principal component analysis (PCA) to investigate the genetic structure in the dataset using TASSEL v.20200220[66]. The genome-wide association studies (GWAS) were then performed by applying a mixed linear model which included the GRM as random factors using the GAPIT v3.0R package[67]. We retrieved false discovery rate (FDR) thresholds for each trait using the p.adjust function implemented in R. The overlap between 5% FDR SNPs and structural variants were assessed with bedtools intersect v2.29.2[68]. We built a contingency table summarizing for each trait the number of 5% FDR SNPs overlapping with indels, translocations, inversions, and duplications. Odds ratios were calculated using two-sided Fisher tests in R.

**Genome-wide sequence feature correlations.** The IPO323 genome was divided into 10 kb bins for which we retrieved sequence-based statistics using the EMBOSS v6.6.0.0 suite and bedtools[68,69]. The dataset included a total of 35 sequence metrics available for the reference genome and the presence of the nine types of

rearrangements identified across the 18 worldwide isolates (summarized in Supplementary Data 5). Population-level single-nucleotide polymorphisms (SNP) were retrieved from 146 *Z. tritici* field isolates[70]. Briefly, sequencing reads were trimmed with trimmomatic (v 0.36) with parameters ILLUMINACLIP:TruSeq3-PE.fa:2:30:10 LEADING:3 TRAILING:3 SLIDING WINDOW:4:15 MINLEN:36 then aligned to the IPO323 reference genome using bowtie2 (–very-sensitive-local; version 2.3.4.3). Variants were called using HaplotypeCaller v.4.0.11.0 and filtered for QUAL > 1000, AN = 20, QD > 5.0, MQ > 20.0, ReadPosRankSum_lower=2.0, ReadPosRankSum_upper=2.0, MQRankSum_lower=2.0, MQRankSum_upper=2.0, BaseQRankSum_lower=2.0, BaseQRankSum_upper=2. Multiallelic SNPs (>2 alleles) were filtered out using bcftools (–norm; version 1.9) and we only retained SNPs with >0.9 genotyping rate and >0.05 allele frequency across the 146 isolates using VCFtools v0.1.15 and bcftools v1.9. The variant call file (VCF) is available from Zenodo (https://zenodo.org/record/4725688)[71]. Data on histone methylation marks (H3K9, H3K4, and H3K27) was previously generated by[72] and included the average read count for enriched domains as identified by RSEG[73]. Recombination rates were calculated using 214 progenies from the cross between isolates 1A5 and 1E4[74] (both included in the pangenome analyses). Pairwise Spearman's correlation values were computed in R using complete observations (*use* = pairwise.complete.obs) and visualized using the *ggcorr* function from the GGally v2.0.0 R package.

**Species-wide model of structural variant occurrence**. We applied a supervised machine-learning approach using 30 sequence features available for all complete genomes with the aim of independently predicting the occurrence of the nine types of structural variation. Full details on the analyzed sequence features are shown in Supplementary Data 5. The training dataset was made of a random 80% subset of the 3,980 10 kb windows. Subsampling different proportions for the train dataset (i.e., 0.2, 0.4, 0.6, and 0.8) and test datasets (i.e., 0.8, 0.6, 0.4 and 0.2, respectively) did not have a meaningful impact on the final model performance (Supplementary Fig. 15). We trained the models for each type of variant separately. To identify the best-performing machine-learning method, we explored four different algorithms based on regression and classification[75–77] (Logistic regression, Stochastic Gradient Boosting, Random Forests, and Boosted Classification Tree methods). Regression and classification methods mostly differ by their procedure to identify decision boundaries. Regression methods implement binary decisions while classification methods (e.g., decision trees) allow for more complex non-linear boundaries[78]. Decision trees can improve the assignment if the two classes follow a non-linear separation, but these methods tend to overfit the training dataset when the two classes are not well separated. All models were trained using the *caret* version 6.0-86 package in R (https://www.rdocumentation.org/packages/caret) with 10-fold cross-validation repeated three times. For the logistic regression models, we used the *glm* method implemented by the caret package using the binomial family definition. For the random forest models, we implemented a random selection of 1 to 10 predictors with the *rf* method (version 4.6-14, including the mtry option). For the boosted classification tree models, we applied the *ada* method (version 2.0-5) using 100, 1000, or 3000 boosting iterations with a maximum tree depth of 1, 5, or 20 and a fixed learning rate of 0.01. For the stochastic gradient boosting models, we used the *gbm* method with 50, 500, or 1000 boosting iterations with a maximum tree depth of 1, 5, or 10. The stochastic gradient boosting models were trained with two values of shrinkage (0.001 or 0.01) and a terminal node minimal size (1 or 5). For all the models, the accuracy measure was used for the selection of the best model during training (metric = "accuracy"). For the final random forest model, we estimated the relative importance of predictors using the *varImp* function from the *caret* package with the scale option set to false.

The eight final models (the 4 different methods using indels and translocations as classifiers, respectively) were tested on a 20% validation subset of the pangenome dataset using the *predict* function. To address the final model performance, we investigated how the true and false positive rate were related using the receiver operating characteristic (ROC) curve implemented in the ROCR R package[79]. The area under the ROC curve (AUC) was calculated with the performance function from ROCR version 1.0-11 package. In addition, we quantified the relationship between the model precision and recall using the precision-recall curves calculated with the *pr.curve* function from the PRROC version 1.3.1 package[80]. Confusion matrices and the evaluation metrics of each model were recovered using the *confusionMatrix* function in the *caret* package.

***Arabidopsis thaliana* pangenome sequence metrics and indel model**. For the *Arabidopsis thaliana* dataset, we used the set of structural rearrangements identified across 8 chromosome-level assemblies[31] (https://1001genomes.org/data/MPIPZ/MPIPZJiao2020/releases/current/). Orthogroups identified by Jiao et al.[31] were defined as core (orthogroup present in all 8 genomes), accessory (orthogroup present in 1–7 genomes), and singletons (orthogroup present in one genome only). Transposable elements and gene annotations were retrieved from the TAIR10 database (https://www.arabidopsis.org/download/index-auto.jsp?dir=%2Fdownload_files%2FGenes%2FTAIR10_genome_release). The list of reference NLRs (Col-0) was retrieved from Weyer et al.[32] (Supplementary Table S3a, annotated genes under the Accession_Name = Col-0_Ref at https://ars.els-cdn.com/content/image/1-s2.0-S0092867419308372-mmc3.xlsx). Transposons

annotation from TAIR10 database were renamed according to the three-letters nomenclature[56] (Supplementary Table 5). The recombination rate in 30 kb windows was estimated from the position of crossover events established by Rowan et al.[9]. We counted the number of crossovers overlapping the sequence windows (Supplementary Data 11). The recombination rate was then calculated as the number of crossovers per 30 kb. GC coverage along the 30 kb windows was calculated using bedtools nuc (v2.29.0). We applied the same four models as for the *Z. tritici* pangenome using sequence metrics gathered along 3974 30 kb windows using the *predict* function in R. Receiver operating characteristic and precision-recall curves were also calculated identically as for the *Z. tritici* pangenome dataset using the ROCR and PRROC R packages (version 1.0-11 and 1.3.1, respectively)[79,80]. The confusion matrices and model evaluation metrics were recovered from the *confusionMatrix* function implemented in the *caret* package.

**Identifying sequence rearrangements across the pedigree**. We applied a similar method applied to the pangenome to call variants in the progenies resulting from the cross between the parental isolates 1A5x1E4[62]. We performed whole-genome alignments of the 9 progeny isolates against the 1A5 parent using nucmer v4.0.0beta2 allowing for multiple matches (–maxmatch option). We filtered the alignments for a minimal identity of 90% and a minimal single match length of 100 bp using the delta-filter in MUMmer v4.0.0beta2. We then extracted the alignment coordinates using show-coords with the -THrd options and mapped the position of syntenic and rearranged regions using the SyRi v1.3 software[24]. Note that the progeny isolate Ztprog08 (M3_08) harbors a fusion between core chromosomes 6 and 12. For accessory chromosomes, the total number of chromosomes included for analysis was lower than for core chromosomes (see Fig. 5C). We focused on rearrangements identified by SyRi including inverted (INV), translocated (TRA), inverted translocated (INVTR), duplicated (DUP), inverted duplicated (INVDP), and highly diverged regions (HDR), together with insertions and deletions considered here as indels (INDEL). Copy gain and loss were considered as copy polymorphisms (CP). Each variant location was given by the breakpoint positions in the 1A5 parental genome. The variant length was calculated as the difference between the two breakpoint positions in the 1A5 parental genome except for insertions, for which we considered the distance between the two breakpoint positions in the progeny genome. For all further analysis, we considered all individual variants larger than 10 bp but shorter than 100,000 bp and removed all variants which were present in the second parent 1E4. We thus retained only the variants generated during the different rounds of meiosis and removed variants existing prior to this. To dissect the emergence and reshuffling of individual rearrangements at each meiotic cycle, we defined four categories based on the variant history in the pedigree (Fig. 5D). Variants present in a focal progeny and the direct parents were defined as retained variants. Variants present in a focal progeny but in none of its ancestors in the pedigree were called new variants. Variants present in a progeny, absent in its direct parents but present already in an ancestor were defined as recurrent variants. Finally, variants absent from a progeny but present in both parents were called lost variants.

**Identification of crossover breakpoints on chromosome 1**. To identify the crossover events on chromosome 1 over the 4 rounds of meiosis, we performed whole-genome alignments of each progeny against its two immediate parents using blastn[44,47] (v2.8.1+). To visualize crossovers, we plotted chromosome-wide synteny using the genoplotR v0.8.9 package in R[81]. To recover the crossover breakpoints, we manually inspected the blastn results (-outputfmt 6) and recovered the approximate positions of shifts in sequence alignment identity between the two parental genomes. We identified a total of 18 crossover breakpoints summarized in Supplementary Table 6. To investigate the relative distance of structural variants to crossover breakpoints and given the overall high synteny in the progeny, we assigned the same breakpoint position to the 1A5 reference genome. We calculated the distance between sequence rearrangements on chromosome 1 and crossovers using the *bedtools closest* command[68,69].

**Evaluation of pangenome sequence rearrangement models for the pedigree**. We applied the four pangenome-trained models to predict indels and translocations to the nine progeny dataset using the *predict* function in the R package *caret* (https://github.com/topepo/caret/). Receiver operating characteristic and precision-recall curves were calculated for the pangenome dataset using the ROCR and PRROC packages[79,80]. Similarly, the confusion matrices and model evaluation metrics were recovered from the *confusionMatrix* function implemented in *caret* package.

**Reporting summary**. Further information on research design is available in the Nature Research Reporting Summary linked to this article.

## Data availability

Genome assemblies for the species-wide pangenome are available at the European Nucleotide Archive (ENA) under the study PRJEB33986 and at https://github.com/crolllab/datasets. The progeny genomes are available at the National Center for Biotechnology Information (NCBI) under the BioProject PRJNA645795. Variant calls for

GWAS were retrieved from the variant call format (VCF) file deposited in the European Nucleotide Archive (ENA) under the accession numbers PRJEB15502/ERP017268 and the analysis number ERZ330467. Variant calls generated for the Swiss field-population[70] were deposited at Zenodo (https://zenodo.org/record/4725688)[71]. Source data are provided with this paper.

## Code availability

The scripts used to perform the synteny analysis, to map the structural rearrangements, and to train the predictive models are available from Zenodo (https://doi.org/10.5281/zenodo.4724074)[71].

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

## Acknowledgements

We are grateful to Leen Abraham, Nikhil Kumar Singh, Sylvain Raffaele, Sam Yeaman, and Aurélien Tellier for critical feedback on a previous version of this manuscript. The sequencing was performed at the Functional Genomics Center Zurich. D.C. was supported by the Fondation Pierre Mercier pour la science and the Swiss National Science Foundation (grant number 31003A_173265).

## Author contributions

T.B. and D.C. designed the study; T.B. performed analyses; S.F. and F.E.H. contributed data sets; M.Z. conceived the progeny pedigree and performed laboratory experiments; T. B. and D.C. wrote the manuscript with input from co-authors.

## Competing interests

The authors declare no competing interests.
