## [Peer Review File · Nature Communications]

REVIEWER COMMENTS

Reviewer #1 (Remarks to the Author):

The work of Badet and colleagues describes structural variation in a fungal pathogen of wheat and then uses a variety of sequence features to develop a machine learning model that predicts the occurrence of structural variation in offspring of specific crosses. The model also appears to be transferable to Arabidopsis plants, suggesting that structural variation may be predictable from genomic features across eukaryotes. This work is important in advancing our understanding of structural variation in fungi and in eukaryotes in general. The manuscript is generally well written and the figures are clear, however, i have a few comments for the authors to consider:

lines 85-86: "The gapless chromosome-level assemblies are among the most complete pangenome of a eukaryotic species." This statement is subjective and may change in the near future as more species / populations are examined. It is also unclear - what does it mean that a pangenome is "complete"? Wouldn't more variation be described as more strains are sequenced? It would be better if the authors actually described the completeness of the pangenome.

line 104: "against the reference genome" - which strain was used as reference? Reading later, i see that different genomes were used as references for different analyses. This is fine, but it would perhaps be best if the authors stated this clearly early on.

line 115: "Overall, 40% of the species gene content is not fixed (i.e accessory genes)". There are two types of variation in gene content - variation due to the presence / absence of accessory chromosomes and variation that's unrelated to the presence / absence of accessory chromosomes. Can the authors distinguish between the two?

lines 121-135: can you add a statement that addresses the possibility / influence of sequence errors on these results?

line 124: "21'717" -> "21,717" - the same issue appears in subsequent numbers so i presume the authors' use of the apostrophe instead of the comma is intentional?

line 126: "less" -> "fewer"

line 139: "High-confidence structural variants" - what does "high confidence" mean?

lines 161-175: Can the authors also describe the percentage of each trait's variation explained by the associated SNPs?

line 194: here you say you used 35 sequence metrics but the abstract and last paragraph of introduction state that you used 30 sequence metrics. Please clarify.

lines 248-250: "The successful application of machine-learning to predict structural variation suggests that eukaryotic genomes rearrange based on clearly identifiable mechanisms." Not sure i fully understand this statement, especially what the authors mean by "clearly identifiable mechanisms". Which are these mechanisms? In my understanding, the power of machine learning approaches is in situations where the data are messy, the mechanisms are unclear, and so on...

Reviewer #2 (Remarks to the Author):

In this study, the authors analyzed the structural variations (SVs) in 19 *Z. tritici* genomes (which were already reported in their previous study), and built machine-learning models with sequence features to predict the SVs across 19 isolates' genomes and 9 pedigree genomes. They also applied these models to *A. thaliana* genomes to predict the events of SV. However, such models are only used to predict the existence of indels and translocations at a large window (10b) in multiple genomes, which means the models only reflect the possible features contributing to the genomic distribution of SV in a population. For a sequenced genome, the accurate positions of the SV cannot be predicted with high resolution. Furthermore, quality controls should be taken to evaluate the effects of different factors (indicated at below) when training the models. Besides, not all types of SV are considered in this study. Therefore, the application of such models will be limited.

Major comments:

According to the definition of Synteny Ratio in the method part "Chromosome-wide synteny ratios", not all pairwise comparisons were considered. This will lead to artifacts as only one reference genome is used. For example, when all other 18 genomes are all syntenic with each other but non-syntenic to the reference genome, the syntenic ratio will be 0. However, the syntenic ratio will be 17/18 when one of other genomes is used as the reference genome.

Considering the high error rates in the PacBio long reads, the read mapping-based SV calling often generate some false SVs and miss a lot of SVs especially duplication, large and complex SVs. As the chromosome-level assemblies of all 19 *Z. tritici* genomes are available, the whole assembly comparisons should give more accurate and comprehensive identification of SVs. For example, the tool SyRI also used by the authors can identify the SVs based on genome sequence alignment.

Line 125-126, considering the high TE content in the genomes, the number of duplications seems too low, which is also different from other studies on eukaryote. Besides, this is also not consistent with the finding in line 196-197, "The highly abundant transposable element superfamily Gypsy shows the strongest correlation with TE regions". The retrotransposons Gypsy are expected to introduce more duplications. Again, the assembly comparison should be good quality control for the accuracy of SV

finding. Besides, it will be more helpful to add information in Table S1 to show which isolate(s) have the SV.

Figure 2D, this figure just means that more SV overlap with secondary metabolites and secreted genes. It is not clear whether the overlapping between the SV and genes are normalized by the total number of these four gene classes.

The author trained a predictive mode of indel and translocation events using the chromosomal sequence features. However, the gene dispensability is not a sequence feature, while itself already mean the existence of structural variation inherently.

The models are trained and applied to predict the existence of SV in 19 genomes. The number and selection of genomes included in the model can affect the polymorphism of SV in the data set, which in turn might affect the accuracy of the models. It would be of interest to see the performance of the models under more tests with different number and selection of genomes. Besides, the models are trained and predicted only for indels and translocations, how their performances on duplications and inversions?

When using the model predict SV in *A. thaliana*, only the indels are predicted. Since other SVs are also available, it would be interesting to see whether these models can be successfully applied to other SVs as claimed at lines 248-250. Besides, why are there still 3993 windows in 120Mb *A. thaliana* genomes like *Z. tritici*?

Again, for the rearrangements identification in 9 progenies, some quality control or validation need to be done because of high error rate in PacBio long reads.

Line 283-284: It is not clear whether the 19 genomes are already sufficient to represent the species-wide set of chromosome rearrangements. And how many SV identified in pedigree genomes are also found in pangenome?

Minor comments:

1. No Figure 3E in Figure 3, please correct it. And what do these different colored links between genomes indicate?
2. Line 210, “we integrated 30 sequence features”, but it seems there are 31 factors included as shown in Table S3.
3. Figure 5E, please clarify the definition of “regions of high identity”. And it’s not clear where these recombination events are.

REVIEWER COMMENTS

Reviewer #1 (Remarks to the Author):

The work of Badet and colleagues describes structural variation in a fungal pathogen of wheat and then uses a variety of sequence features to develop a machine learning model that predicts the occurrence of structural variation in offspring of specific crosses. The model also appears to be transferable to Arabidopsis plants, suggesting that structural variation may be predictable from genomic features across eukaryotes. This work is important in advancing our understanding of structural variation in fungi and in eukaryotes in general. The manuscript is generally well written and the figures are clear, however, i have a few comments for the authors to consider:

lines 85-86: "The gapless chromosome-level assemblies are among the most complete pangenome of a eukaryotic species." This statement is subjective and may change in the near future as more species / populations are examined. It is also unclear - what does it mean that a pangenome is "complete"? Wouldn't more variation be described as more strains are sequenced? It would be better if the authors actually described the completeness of the pangenome.

RESPONSE: We fully agree that this statement may only have temporary meaning. We have now revised the sentence to state more explicitly that this species' pangenome was built by covering the global distribution range and is based on telomere-to-telomere assemblies.

line 104: "against the reference genome" - which strain was used as reference? Reading later, i see that different genomes were used as references for different analyses. This is fine, but it would perhaps be best if the authors stated this clearly early on.

RESPONSE: Yes, this was unclear. In response also to the other reviewer, we have added an entirely new approach how to assess structural variation. This new approach is no longer dependent on referencing any particular genome. We have adjusted the sentence to reflect the new approach and clarify our procedure.

line 115: "Overall, 40% of the species gene content is not fixed (i.e accessory genes)". There are two types of variation in gene content - variation due to the presence / absence of accessory chromosomes and variation that's unrelated to the presence / absence of accessory chromosomes. Can the authors distinguish between the two?

RESPONSE: Yes, it would be possible to assess the degree of gene presence/absence variation associated with whole-chromosome presence/absence variation (of accessory chromosomes). However, all accessory chromosomes combined account for only ~5% of the total gene content. Hence, the vast majority of the accessory gene presence/absence

polymorphism is due variation in core chromosomes. For this reason, we decided to focus on rearrangements affecting segments of chromosomal sequences without considering the presence or absence of entire chromosomes.

lines 121-135: can you add a statement that addresses the possibility / influence of sequence errors on these results?

RESPONSE: Sequencing errors are a clear concern and we implemented approaches, which should be robust overall. But we agree that the concern should be explicitly mentioned and do this now at the requested line. In fact, we implemented an alternative variant calling procedure based on whole-genome alignments (SyRi), bypassing the potential errors from uncorrected read mapping. We performed all subsequent analyses based on variants called with the new whole-genome alignment method.

line 124: "21'717" -> "21,717" - the same issue appears in subsequent numbers so i presume the authors' use of the apostrophe instead of the comma is intentional?

RESPONSE: We have changed the format. We have accidentally used the Swiss format for thousand separators.

line 126: "less" -> "fewer"

RESPONSE: Corrected.

line 139: "High-confidence structural variants" - what does "high confidence" mean?

RESPONSE: We have removed the term as this was unclear. The variant calling approach has also been changed and hence the corresponding sentence.

lines 161-175: Can the authors also describe the percentage of each trait's variation explained by the associated SNPs?

RESPONSE: Yes, we have added this information now to Table S4 and also indicate this in the text: "The genetic diversity sampled across the 106 isolates explained between 0 and 26% of the phenotypic trait variation (for growth at 14 days after treatment with propiconazole and growth at 22°C after 14 days respectively; Table S4)"

line 194: here you say you used 35 sequence metrics but the abstract and last paragraph of introduction state that you used 30 sequence metrics. Please clarify.

RESPONSE: The correlation analysis was performed using 35 different metrics including three types of histone marks, SNP counts and gene singletons. The latter five metrics were not available for the progeny pedigree. We state this now more clearly throughout the text.

lines 248-250: "The successful application of machine-learning to predict structural variation suggests that eukaryotic genomes rearrange based on clearly identifiable mechanisms." Not sure i fully understand this statement, especially what the authors mean by "clearly identifiable mechanisms". Which are these mechanisms? In my understanding, the power of machine learning approaches is in situations where the data are messy, the mechanisms are unclear, and so on...

RESPONSE: The wording was indeed unclear. We have now rephrased the sentence to: The successful application of machine-learning to predict segregating structural variation suggests that eukaryotic genomes rearrange based on identifiable patterns in genome sequences.

#####

Reviewer #2 (Remarks to the Author):

In this study, the authors analyzed the structural variations (SVs) in 19 Z. tritici genomes (which were already reported in their previous study), and built machine-learning models with sequence features to predict the SVs across 19 isolates' genomes and 9 pedigree genomes. They also applied these models to A. thaliana genomes to predict the events of SV. However, such models are only used to predict the existence of indels and translocations at a large window (10b) in multiple genomes, which means the models only reflect the possible features contributing to the genomic distribution of SV in a population. For a sequenced genome, the accurate positions of the SV cannot be predicted with high resolution. Furthermore, quality controls should be taken to evaluate the effects of different factors (indicated at below) when training the models. Besides, not all types of SV are considered in this study. Therefore, the application of such models will be limited.

Major comments:

According to the definition of Synteny Ratio in the method part "Chromosome-wide synteny ratios", not all pairwise comparisons were considered. This will lead to artifacts as only one reference genome is used. For example, when all other 18 genomes are all syntenic with each other but non-syntenic to the reference genome, the syntenic ratio will be 0. However, the syntenic ratio will be 17/18 when one of other genomes is used as the reference genome.

RESPONSE: We entirely agree that our original approach could have caused artefacts to some degree. We have now revised the procedure and calculate synteny ratios based on all-against-all pairwise comparisons in 10 kb bins. This new approach eliminates the above identified issue. We now use this newly calculated metric onward in the manuscript.

Considering the high error rates in the PacBio long reads, the read mapping-based SV calling often generate some false SVs and miss a lot of SVs especially duplication, large and complex SVs. As the chromosome-level assemblies of all 19 Z.tritici genomes are available, the whole assembly comparisons should give more accurate and comprehensive identification of SVs. For example, the tool SyRI also used by the authors can identify the SVs

based on genome sequence alignment.

RESPONSE: Yes, we fully agree that mapping of noisy reads to reference genome can lead to erroneous SV calls and to a reference-bias as identified above. Following the reviewer's recommendations, we have now entirely replaced the SV calling procedure and use whole-genome alignments (*i.e.* SyRI), overcoming the shortcomings of the read mapping approach. In the manuscript, we now briefly compare the outcomes of the two SV calling procedures and show an overall convergence in the detected SVs. However, the whole-genome alignment method enabled us to resolve more complex rearrangements (as suspected by the reviewer). Based on the significant improvement in the reliability and precision of the new procedure, we have re-performed all subsequent analyses to integrate the new procedure. We believe that this not only allows more accurate but also more generalizable insights into the extent of structural variation.

Line 125-126, considering the high TE content in the genomes, the number of duplications seems too low, which is also different from other studies on eukaryote. Besides, this is also not consistent with the finding in line 196-197, "The highly abundant transposable element superfamily Gypsy shows the strongest correlation with TE regions". The retrotransposons Gypsy are expected to introduce more duplications. Again, the assembly comparison should be good quality control for the accuracy of SV finding. Besides, it will be more helpful to add information in Table S1 to show which isolate(s) have the SV.

RESPONSE: As suggested and described above, we implemented a new SV calling procedure, which indeed resulted in a higher number of resolved duplication events jointly with a decrease in translocations detected among genomes. We now also provide isolate-specific variant information in Table S1 and S2 as requested.

Figure 2D, this figure just means that more SV overlap with secondary metabolites and secreted genes. It is not clear whether the overlapping between the SV and genes are normalized by the total number of these four gene classes.

RESPONSE: Yes, Figure 2D displays the proportion of genes in each category that overlap with structural variants. We clarified this in the legend.

The author trained a predictive model of indel and translocation events using the chromosomal sequence features. However, the gene dispensability is not a sequence feature, while itself already mean the existence of structural variation inherently.

RESPONSE: We agree that gene dispensability does not represent a sequence feature *per se* and is possibly directly tied to a structural variation (*i.e.* a gene deletion). However, gene dispensability in the study species is mostly associated with loss-of-function mutations in the coding sequence (Plissonneau et al 2018, BMC Biology). Hence, we see value in including this characteristic of sequence windows along the genome in order to find correlates with specific types of SVs. We state now more explicitly though the particular nature of "gene dispensability".

The models are trained and applied to predict the existence of SV in 19 genomes. The number and selection of genomes included in the model can affect the polymorphism of SV in the data set, which in turn might affect the accuracy of the models. It would be of interest to see the performance of the models under more tests with different number and selection of genomes.

RESPONSE: This is a very interesting suggestion. We trained now ten models using different fractions of the total pangenome dataset (0.2, 0.4, 0.6 and 0.8) and computed performance metrics using the rest of the dataset (*i.e.* 0.8, 0.6, 0.4 and 0.2 respectively). We present the results now in Figure S15. It is interesting to note that the size of the training dataset has a fairly small impact on the accuracy of the models.

Besides, the models are trained and predicted only for indels and translocations, how their performances on duplications and inversions?

RESPONSE: As suggested, we have implemented now an alternative SV calling procedure based on whole-genome alignments (SyRi). With this, we are able to resolve more complex variants, including duplications, inverted duplications, inverted translocations, copy number variants, tandem repeats and highly diverged regions. We trained models for each of the nine types of variants and present the results in Figures 4, S6, S7 and Table S7, S8 and S9. The four different algorithms performed similarly on all variant types. Best results were achieved for indels and highly diverged regions. Models for inversions and tandem duplications, the two rarest types of variants, yielded high accuracy (due to strong true negative detection) but had low precision and recall (no true positives detected). We mention this now in the relevant results section.

*When using the model predict SV in *A. thaliana*, only the indels are predicted. Since other SVs are also available, it would be interesting to see whether these models can be successfully applied to other SVs as claimed at lines 248-250.*

RESPONSE: Following the previous recommendation, we also replaced the SV calling procedure to resolve more complex variants, including duplications, inverted duplications, inverted translocations, copy number variants, tandem repeats and highly diverged regions in the *A. thaliana* genome. We trained a model for each of the nine SV types (Table S10-11). We find that: we correctly predicted 26% of the highly diverged regions, 20% of the duplications and copy number variation, 17% of the inverted duplications and translocations and 16% of the inverted translocations. The models were not able to predict the two rarest variants in *Z. tritici* and *A. thaliana* genomes, namely inversions and tandem duplications (6/182 and 39/963 events respectively).

*Besides, why are there still 3993 windows in 120Mb *A. thaliana* genomes like *Z. tritici*?*

RESPONSE: To account for the difference in genome size between the two species, we splitted the genome of *A. thaliana* into 30 kb bins. This allowed us to perform model predictions on a similar number of genomic windows for more direct performance comparisons.

Again, for the rearrangements identification in 9 progenies, some quality control or validation need to be done because of high error rate in PacBio long reads.

RESPONSE: We have also performed the new SV calling based on whole-genome alignments (SyRI) for the progeny dataset (including model predictions). This new approach, as indicated already by the reviewer, circumvents issues of PacBio long read sequencing errors. Given the higher confidence in the SyRI approach, we have replaced all results from progeny genomes with the new SV call procedure. Hence, the manuscript now relies entirely on the more robust SV calling procedure. We mention the read mapping approach only at the beginning to transparently show variation in the outcomes.

Line 283-284: It is not clear whether the 19 genomes are already sufficient to represent the species-wide set of chromosome rearrangements.

RESPONSE: We fully agree with this concern. We have performed now a resampling analysis by selecting variable subsets of genomes and report the total number of discovered SVs (SyRI approach) per set of genomes. The results are shown in Figure S2. We find that we have not yet reached saturation of total SVs (using both read mapping and whole-genome alignment methods). It is important to note that *Z. tritici* is among the most polymorphic, fully sexual eukaryotic species analyzed to date. We believe that the geographic breadth of the 19 complete genomes covering all major areas of occurrence provide a reasonable picture of the total diversity. Furthermore, we are showing now the power of the predictive model based on subsets of the total genome set (see above) and find that performance is not meaningfully impacted by different sets and numbers of genomes. We now more critically mention all these points in the text though.

And how many SV identified in pedigree genomes are also found in pangenome?

RESPONSE: This is an excellent suggestion. We have now called the progeny variants using the whole-genome alignment method and the IPO323 genome as a reference to compare SVs. We find that 189 newly emerged variants in the progeny are also found in the pangenome SV set. This suggests that a few rounds of meiosis are sufficient to re-generate variants already segregating in the species. We mention this now in the progeny section of the results.

Minor comments:

1. No Figure 3E in Figure 3, please correct it. And what do these different colored links between genomes indicate?

RESPONSE: Error corrected. We now also state that darker colored links represented higher sequence identity.

2. Line 210, “we integrated 30 sequence features”, but it seems there are 31 factors included

as shown in Table S3.

RESPONSE: The coverage in singleton genes was indeed not considered for model training. We adjusted this now accordingly.

3. Figure 5E, please clarify the definition of “regions of high identity”. And it’s not clear where these recombination events are.

RESPONSE: We have updated Figure 5E to clarify the location of the recombination events and state now that regions of maximal pairwise identity (100%) are colored in blue.

REVIEWERS' COMMENTS

Reviewer #1 (Remarks to the Author):

The authors have satisfactorily addressed my concerns.

Reviewer #2 (Remarks to the Author):

The authors have addressed my concerns in the revision. Only some minor comments:

Line 179: "Consistent with this, we find that inversions are more likely to affect coding sequences". Here the "affect", do you mean the genes are inside or at breakpoints of the inversions?

Line 199: "Significantly associated SNPs were over-represented near inversions". Here, the "near" is not clearly defined, and also for the "near" at the line 232, line 235, line 236

Line 204: Figure 3F-G. G is not in the figure.